# Mind the gap: difference between Framingham heart age and real age increases with age in HIV-positive individuals–a clinical cohort study

Teri-Louise Davies,[1] Mark Gompels,[2] Sarah Johnston,[2] Begoña Bovill,[2] Margaret T May[1]

This work was presented in part at the Eleventh International Congress on Drug Therapy in HIV Infection Glasgow, UK 11–15 November 2012. Abstract available: Davies T et al Journal of the International AIDS Society 2012, 15(Suppl 4):18152. http://www.jiasociety.org/index.php/jias/article/view/18152 | http://dx.doi.org/10.7448/IAS.15.6.18152.

For numbered affiliations see end of article.

**Correspondence to**
Dr Margaret May;
m.t.may@bristol.ac.uk

## ABSTRACT

**Objectives:** To measure the excess risk of cardiovascular disease (CVD) in HIV-positive individuals by comparing 'heart age' with real age and to estimate associations of patients' characteristics with heart age deviation (heart age–real age).

**Design:** Clinical Cohort Study.

**Setting:** Bristol HIV clinic, Brecon Unit at Southmead Hospital, Bristol, UK.

**Participants:** 749 HIV-positive adults who attended for care between 2008 and 2011. Median age was 42 years (IQR 35–49), 67% were male and 82% were treated with antiretroviral therapy.

**Main outcome measures:** We calculated the Framingham 10-year risk of CVD and traced back to 'heart age', the age of an individual with the same score but ideal risk factor values. We estimated the relationship between heart age deviation and real age using fractional polynomial regression. We estimated crude and mutually adjusted associations of sex, age, CD4 count, viral load/treatment status and period of starting antiretroviral therapy with heart age deviation.

**Results:** The average heart age for a male aged 45 years was 48 years for a non-smoker and 60 years for a smoker. Heart age deviation increased with real age and at younger ages was smaller for females than males, although this reversed after 48 years. Compared to patients with CD4 count <500 cells/mm$^3$, heart age deviation was 2.4 (95% CI 0.7 to 4.0) and 4.3 (2.3 to 6.3) years higher for those with CD4 500–749 cells/mm$^3$ and ≥750 cells/mm$^3$, respectively.

**Conclusions:** In HIV-positive individuals, the difference between heart age and real age increased with age and CD4 count and was very dependent on smoking status. Heart age could be a useful tool to communicate CVD risk to patients and the benefits of stopping smoking.

## INTRODUCTION

HIV-positive individuals are now living to much older ages[1][2] and therefore may be at high risk of cardiovascular disease (CVD).[3][4]

## ARTICLE SUMMARY

### Strengths and limitations of this study

- We were able to estimate Framingham risk and heart age in three-quarters of the patients in the Bristol Cohort.
- Some patient groups, such as those with a history of injection drug use, were underrepresented and therefore our results may be more applicable to patients that regularly attend HIV care.
- We did not have information on smoking status. We sought to overcome this by duplicating analyses assuming all were smokers and all were non-smokers.
- We lacked information on cardiovascular disease (CVD) events or deaths and so we do not know whether higher estimated CVD risk in this HIV population translates to an elevated rate of CVD.

Guidelines for the clinical management of HIV patients stress the importance of assessing risk of CVD and recommend interventions to treat risk factors.[5] Although there have been some attempts to introduce CVD risk scoring tools specifically for HIV-positive individuals,[6–8] none have been independently validated and therefore the Framingham risk equation[9] is still widely used,[10] particularly as the Framingham Heart Study website[11] provides a simple, accessible tool for calculating the risk of developing CVD within 10 years.

Communication of CVD risk to HIV patients is extremely important, particularly the impact of modifiable risk factors, such as smoking. Recently the Canadian Cardiovascular Society[12] promoted the use of 'heart age' derived from the 10-year Framingham risk equations for general CVD.[9] A person's 'heart age' is the age of an individual with the same risk score but ideal modifiable risk factor values. Therefore 'heart age' is a useful measure of excess CVD risk adjusted for age and sex.

Our objectives were to compare 'heart age' with actual age and to estimate the association of patient clinical and demographic characteristics with heart age deviation, the difference between estimated 'heart age' and their real age, in the Bristol Cohort of HIV-positive individuals.

## METHODS
### Study participants
The Bristol HIV Cohort study enrols patients attending the Brecon Unit at Southmead Hospital, Bristol, UK. Routine clinical data collected on patients attending for HIV care up to November 2011 were available for analysis as part of the UK CHIC study. In accordance with data protection policy, all data were anonymised. Included patients were aged 18 years and over and were not infected perinatally.

### Data measurement and availability
Demographic data on sex, date of birth, ethnicity (black African, white and other), assumed HIV transmission group and the dates of HIV diagnosis and first clinic visit were available. CD4 cell count and HIV-1 RNA were usually measured at each clinic visit. Details of antiretroviral therapy (ART) and non-HIV medications were available. Systolic blood pressure (SBP) and total and high-density lipoprotein (HDL) cholesterol have been measured since 2008, at first visit and at least annually thereafter according to protocol. Patients ever recorded as taking antihypertensive medication were classed as treated for high blood pressure. Diagnosis of diabetes mellitus was recorded in patient notes. Patients with missing diabetes status were assumed to be non-diabetic. Smoking status was not available. Patients included in analyses had at least one set of Framingham risk factors (SBP, total and HDL cholesterol), CD4 count and HIV-1 RNA measured within a 6-month time window. We used the latest available measurements to calculate the Framingham risk score.

### Statistical methods
#### Calculation of Framingham 10-year risk of CVD and heart age
We calculated the 10-year risk of CVD for each person using the sex-specific Framingham equations for general CVD[9][11] which include age, total and HDL cholesterol, SBP, treatment for hypertension, current smoking (yes/no) and diabetes status. We used the Framingham risk to trace back to 'heart age': the age of an individual with the same score but ideal risk factor values (non-smoker, non-diabetic, untreated SBP 125 mm Hg, total cholesterol 180 mg/dL, HDL 45 mg/dL[9]). For example, if a 40-year-old man has a 10-year CVD risk of 5.6%, his 'heart age' would be 45 years because a 45-year-old man with ideal risk factors has a 10-year risk of CVD of 5.6%. For comparison, the 10-year CVD risk for a 40-year-old man with ideal risk factors is 3.9%. Data on smoking

status was not collected in the Bristol cohort and therefore analyses were conducted twice, first assuming all were smokers, and second assuming all were non-smokers. Heart age deviation ('heart age'–real age) was calculated for each individual for each smoking assumption.

### Analysis of heart age deviation
We estimated the difference between age and 'heart age' overall and by age group (18–39, 40–49, 50–59 and ≥60). We used box plots to compare the distribution of 'heart age' with median real age for male and female smokers and non-smokers stratified by real age group. We used fractional polynomial regression models[13] separately for male and female smokers and non-smokers to show the variation of heart age deviation with age. We used univariable and multivariable linear regression models to estimate crude and mutually adjusted associations of sex, age group, current CD4 count, treatment/viral load status (untreated, treated and suppressed, treated and not suppressed) and period of starting ART (pre vs post 2003) with heart age deviation. We also considered models that included duration since HIV diagnosis, duration since first clinic visit, duration and type of ART.

In sensitivity analyses, we repeated the main analyses first including only those on ART and second restricting to men. We also tested whether CD4 count was associated with total cholesterol, the ratio of total to HDL cholesterol and SBP, controlling for age and sex. Results are presented as the difference between heart age and real age in years with 95% CIs. All calculations were executed in STATA V.12.1.[14]

## RESULTS
Of the 1013 patients who attended the clinic, 749 (74%) had measurements of CD4 count, HIV-1 RNA, total and HDL cholesterol, SBP taken within a 6-month period. Patient demographic and clinical characteristics at the time of the Framingham risk assessment are shown in table 1. Two-thirds of the patients were men, the majority of whom were men who have sex with men (MSM) and of white ethnicity. In contrast, the majority of women (63%) were of black African origin. Compared to those included in the analyses, excluded individuals (N=264) without risk factor measurements were similar in age, sex and ethnicity, but were twice as likely to be infected through injection drug use (IDU), blood product or 'unknown' risk group. The median latest CD4 count was 484 (IQR 322–657) mm³ in those excluded which was intermediate between the treated and untreated included patients (table 1).

Table 2 shows real age compared with heart age and estimated heart age deviation overall and by age group, stratified by sex and smoking status. Heart age was greater than real age for all groups except for non-smoking women aged 18–39 years. The mean age of

**Table 1** Patient demographic and clinical characteristics at time of Framingham Score assessment

| | Number of patients N=749 | Per cent |
|---|---|---|
| **Sex** | | |
| Male | 503 | 67.2 |
| **Transmission risk group** | | |
| Heterosexual | 386 | 51.5 |
| MSM | 312 | 41.7 |
| IDU | 23 | 3.1 |
| Blood product | 7 | 0.9 |
| Unknown | 21 | 2.8 |
| **Ethnicity** | | |
| Black African | 228 | 30.4 |
| White | 441 | 58.9 |
| Other | 80 | 10.7 |
| **Age (years) (median, IQR)** | 42.2 | (35.5–49.4) |
| **Age category (years)** | | |
| 18–39 | 310 | 41.4 |
| 40–49 | 265 | 35.4 |
| 50–59 | 118 | 15.8 |
| 60+ | 56 | 7.5 |
| **Treatment status** | | |
| On ART | 612 | 81.7 |
| **Started ART pre 01/01/2003** | 168 | 22.4 |
| **Viral load (copies per ml)** | | |
| Treated and suppressed (vl≤50) | 535 | 71.4 |
| Treated and unsuppressed | 77 | 10.3 |
| Untreated | 137 | 18.3 |
| **Median CD4 count (mm$^3$), untreated (IQR)** | 477 | (334–602) |
| **Median CD4 count (mm$^3$), treated (IQR)** | 559.5 | (409–721) |
| **CD4 count (mm$^3$)** | | |
| <500 | 318 | 42.5 |
| 500–749 | 275 | 36.7 |
| ≥750 | 156 | 20.8 |
| **Median total cholesterol(mg/dL) (IQR)** | 189.1 | (162–220) |
| **Median HDL cholesterol(mg/dL) (IQR)** | 50.2 | (38.6–61.8) |
| **Median systolic blood pressure(mmHg) (IQR)** | 131 | (117–143) |

ART, antiretroviral therapy; HDL, high-density lipoprotein; IDU,injection drug use; IQR, interquartile range; MSM, men who have sex with men.

men was 44.3 years and their mean heart age was 47.7 years if a non-smoker or 59.2 years if a smoker. The women were slightly younger with a mean age of 40.6 years and corresponding mean heart ages of 42.4 and 53.1 years assuming non-smoker and smoker, respectively. Figure 1 illustrates the distribution of heart age by age group. The box represents the middle 50% of the distribution with the median marked as a line, and the tails extend to 95% of the distribution with outliers marked as dots. There was an increasing trend across age groups in the deviation between the median age and median heart age.

Figure 2 shows heart age deviation increased substantially with real age and was much higher for smokers, for example, for men aged 50, the heart age deviation

was around 18 years in smokers and around 5 years in non-smokers (illustrated by green line). At younger ages, women had smaller heart age deviation than men, but this reversed after about age 48. However, the deviation is relative to sex-specific risk in those with ideal risk factors and women have lower absolute risk than men in the general population. In our study, prevalence of diabetes increased with age, as expected. Median total and HDL cholesterol were higher in men aged ≥40 years compared with younger men, and were also higher in women aged ≥50 years compared with younger women.

The crude and adjusted associations of variables with heart age deviation are shown in table 3. Duration since HIV diagnosis, duration since first clinic visit and duration of ART were not associated with heart age

**Table 2** Age, heart age and heart age deviation (heart age–real age) by age group for male and female smokers and non-smokers

| Sex | Smoking assumption | Real age category (years) | n (%) | Mean real age (years) | Mean heart age (years) | Deviation=heart age-real age (years) | |
|---|---|---|---|---|---|---|---|
| | | | | | | Mean | (95% CI) |
| Males | Non-smoker | 18–39 | 184 (37) | 33.5 | 34.9 | 1.3 | (0.60 to 2.09) |
| | | 40–49 | 182 (36) | 45.1 | 48.5 | 3.3 | (2.17 to 4.53) |
| | | 50–59 | 91 (18) | 53.4 | 59.5 | 6.1 | (3.85 to 8.37) |
| | | 60- | 46 (9) | 66.2 | 73.0 | 6.8 | (2.19 to 11.36) |
| | | Total | 503 (100) | 44.3 | 47.7 | 3.4 | (2.65 to 4.21) |
| Males | Smoker | 18–39 | 184 (37) | 33.5 | 43.3 | 9.8 | (8.78 to 10.74) |
| | | 40–49 | 182 (36) | 45.1 | 60.1 | 15.0 | (13.53 to 16.48) |
| | | 50–59 | 91 (18) | 53.4 | 73.9 | 20.4 | (17.64 to 23.24) |
| | | 60- | 46 (9) | 66.2 | 89.6 | 23.3 | (18.35 to 28.29) |
| | | Total | 503 (100) | 44.3 | 59.2 | 14.8 | (13.82 to 15.84) |
| Females | Non-smoker | 18–39 | 126 (51) | 33.1 | 31.7 | -1.4 | (-2.93 to 0.05) |
| | | 40–49 | 83 (34) | 44.1 | 47.1 | 3.0 | (-0.13 to 6.10) |
| | | 50–59 | 27 (11) | 55.2 | 62.9 | 7.7 | (1.54 to 13.93) |
| | | 60- | 10 (4) | 65.9 | 82.2 | 16.3 | (0.07 to 32.52) |
| | | total | 246 (100) | 40.6 | 42.4 | 1.8 | (0.15 to 3.41) |
| Females | Smoker | 18–39 | 126 (51) | 33.1 | 39.9 | 6.8 | (4.87 to 8.64) |
| | | 40–49 | 83 (34) | 44.1 | 59.2 | 15.0 | (11.19 to 18.90) |
| | | 50–59 | 27 (11) | 55.2 | 78.9 | 23.7 | (15.77 to 31.55) |
| | | 60- | 10 (4) | 65.9 | 99.8 | 33.9 | (16.79 to 51.01) |
| | | total | 246 (100) | 40.6 | 53.1 | 12.5 | (10.42 to 14.61) |

deviation. Compared with ages 18–39, those aged ≥60 years had an increase of 8.87 (95% CI 5.90 to 11.84) years in heart age deviation, and this was approximately doubled in smokers. Compared with those with CD4 count <500 cells/mm$^3$, those with CD4 count ≥750 cells/mm$^3$ had an increase of 4.28 (95% CI 2.28 to 6.27) years in heart age deviation and this effect was independent of smoking status. When analyses were

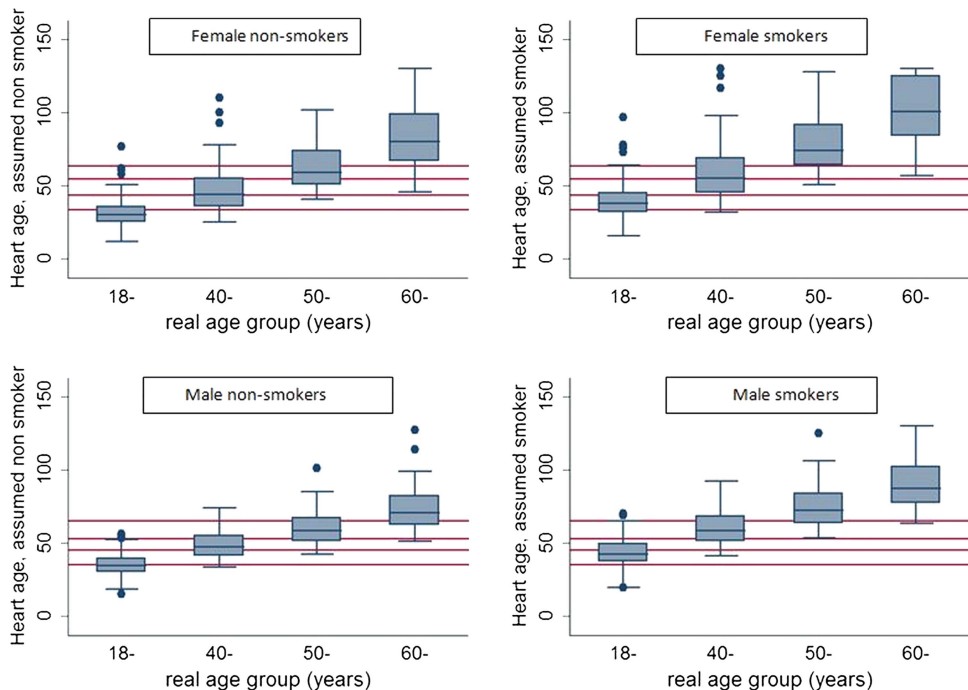

**Figure 1** Box plots of distribution of heart age by age group for female non-smokers (upper left), female smokers (upper right), male non-smokers (lower left) and male smokers (lower right). The median real age for each age group is shown in red for comparison. In the box plot, the whiskers include 95% of the distribution and the box includes the middle 50% with the median marked as a line. Outliers are shown as filled circles.

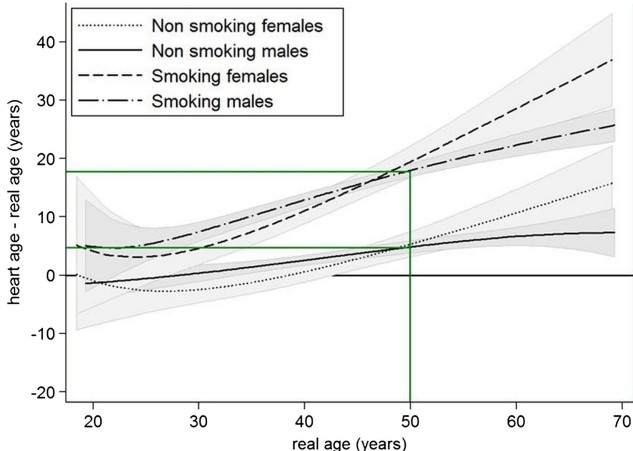

**Figure 2** Heart age deviation in years (95% CI shown shaded in gray) for male and female smokers and non-smokers for real ages 20–70 years.

restricted to men only, the associations of age and CD4 count with heart age deviation were somewhat weaker. Patterns of results were similar when analyses were restricted to individuals on ART. There was no evidence of a difference in heart age deviation between those on protease inhibitor (PI) compared with non-nucleoside reverse transcriptase inhibitor (NNRTI)-based ART at the time of the CVD risk assessment.

Higher total cholesterol (adjusted for age and sex) was associated with higher CD4 count (80 mg/dL increase in cholesterol per 50 increase in CD4 count,

p=0.01) and this was not attenuated by adjusting for HDL cholesterol. Higher total:HDL cholesterol was similarly associated with higher CD4 count. SBP was not associated with CD4 count. There was some evidence that treated patients with unsuppressed virus, but not untreated patients, had greater heart deviation than those with suppressed virus.

## DISCUSSION
### Main results
We showed that in the Bristol HIV cohort on average 'heart age' was greater than real age for men of all ages and for women aged over 40. Heart age deviation widened with increasing age and was very much higher if people smoke. On average, a 45-year-old male smoker had a 'heart age' of around 60 years. Our results suggest that in women the difference between age and 'heart age' increased steeply after menopausal age. Untreated patients and treated patients who were virally suppressed had similar heart ages, but those who were not virally suppressed on ART had higher heart age deviation. Higher CD4 count was associated with higher heart age deviation.

### Strengths and limitations
As far as we are aware, this is the first study to calculate 'heart age' based on the Framingham CVD risk score for HIV-positive individuals. Complete data on Framingham risk factors and HIV biomarkers were only

**Table 3** Crude and mutually adjusted heart age deviation (difference from comparator group*) according to patient characteristics for non-smokers and smokers

| Variable | All individuals (N=749) | | | |
| | Non smoker deviation (years) (95% CI) | | Smoker deviation (years) (95% CI) | |
| | Crude | Adjusted | Crude | Adjusted |
|---|---|---|---|---|
| Female vs male | −1.65 (−3.25 to −0.05) | −0.96 (−2.52 to 0.59) | −2.32 (−4.38 to −0.25) | −0.72 (−2.62 to 1.18) |
| Age category | – | – | – | – |
| 18–39* | – | – | – | – |
| 40–49 | 3.02 (1.35 to 4.69) | 3.38 (1.66 to 5.10) | 6.48 (4.44 to 8.52) | 6.75 (4.65 to 8.86) |
| 50–59 | 6.27 (4.11 to 8.43) | 6.39 (4.17 to 8.61) | 12.64 (10.00 to 15.27) | 12.64 (9.94 to 15.35) |
| 60+ | 8.26 (5.36 to 11.16) | 8.87 (5.90 to 11.84) | 16.67 (13.13 to 20.20) | 17.21 (13.59 to 20.83) |
| CD4 count category (cells/mm³) | – | – | – | – |
| <500* | – | – | – | – |
| 500–750 | 1.91 (0.23 to 3.60) | 2.39 (0.74 to 4.03) | 2.50 (0.34 to 4.66) | 3.01 (1.00 to 5.02) |
| ≥750 | 4.23 (2.23 to 6.23) | 4.28 (2.28 to 6.27) | 6.12 (3.55 to 8.68) | 5.44 (3.01 to 7.87) |
| Viral load category | – | – | – | – |
| Treated with suppressed vl* | – | – | – | – |
| Treated with unsuppressed vl | 1.07 (−1.44 to 3.59) | 2.84 (0.40 to 5.29) | −0.19 (−3.42 to 3.04) | 2.90 (−0.08 to 5.89) |
| Untreated | −1.39 (−3.36 to 0.58) | 0.27 (−1.74 to 2.27) | −3.61 (−6.15 to −1.07) | −0.28 (−2.73 to 2.17) |
| ART start date pre 01/01/2003 | 0.18 (−1.62 to 1.99) | −2.28 (−4.14 to −0.42) | 1.98 (−0.35 to 4.30) | −2.43 (−4.70 to −0.17) |
| Constant | – | −1.26 (−3.00 to 0.49) | – | 6.70 (4.57 to 8.82) |

ART, antiretroviral therapy.

available on 74% of the patients in the Bristol Cohort and some patient groups such as IDU were under-represented. Therefore, our results may be more applicable to patients that regularly attend for HIV care. Measurement error and misclassification may have biased our results. We only used one cross-sectional assessment of Framingham CVD risk score and cholesterol measurements were not all fasting measures. We assumed that individuals without diabetes status recorded were not diabetic and that those who had ever been prescribed medication for high blood pressure remained treated which may have resulted in some misclassification. A major limitation of our study is that we did not have information on smoking status. We sought to overcome this by duplicating analyses assuming all were smokers and all were non-smokers. Our results may be biased if smoking is associated with changes in other risk factors such as SBP and cholesterol that are intermediate in the pathway from smoking to CVD. Our study was limited to cross-sectional analysis and therefore does not show within-person changes in CVD risk. We do not yet have full information on CVD events or deaths and so we do not know whether higher estimated CVD risk in this HIV population translates to an elevated rate of CVD. A much larger number of patients would be required to properly analyse CVD events and causes of death.

The calculation of 'heart age' uses a set of 'ideal' risk factor values proposed by the Framingham investigators. It is likely that 'normal' or 'average' CVD risk factor values in the UK general population are somewhat worse than the 'ideal'. According to the Health Survey for England 2006 which reported CVD risk factors in adults, for men aged >35 years the mean total and HDL cholesterol were 220 and 50 mg/dL, respectively, somewhat higher than the ideal values of 180 and 45 mg/dL, and only 69% had SBP<140 and DBP<90 mm Hg without medication.[15] Unfortunately, we did not have an age and sex matched HIV-negative population for direct comparison.

### In context with other studies

Our finding that heart age was greater than real age for the majority of HIV-positive individuals in this study population is concordant with a study that found increased CVD rates among HIV-positive compared with negative controls.[16] The increased burden of CVD among HIV-positive individuals is likely a consequence of increased traditional risk factors, including dyslipidemia and insulin resistance and non-traditional risk factors such as immune activation and inflammation that may contribute to the accelerated ageing process characterised by higher than expected rates of non-infectious comorbidities.[17] Higher prevalence of smoking also contributes to the CVD epidemic in the HIV-positive population[18 19] as may the use of recreational drugs. ART itself may contribute to CVD, but no effect of current PI use was detected in this UK cohort.

We found that heart age deviation was greater at older ages. This is particularly significant as the mean age of HIV infection is increasing and it is predicted that by 2020 in the USA 50% of people living with HIV will be over 50 years old.[20] Although our study was cross-sectional, it is likely that the gap between real and heart age increases within individuals as they age. Older HIV-positive individuals, compared with matched controls, have been found to have a higher prevalence of hypertension, hypertriglyceridaemia, low bone density and lipodystrophy suggesting that HIV and treatment-related factors accelerate normal ageing.[3]

However, in our study, the widening of the gap between heart age and real age seen in older patients must be entirely driven by the risk factors included in the Framingham equation, namely diabetes, SBP, total and HDL cholesterol, being worse at older ages.

Our finding that higher current CD4 count was associated with higher estimated Framingham risk of CVD is to be interpreted with caution since the SMART study of structured treatment interruption based on CD4 count found that CVD events were greater in those with lower CD4.[21] However, our results are in line with another study that found a higher prevalence of clinically evident lipodystrophy and higher CD4 cell counts in patients with higher Framingham scores.[22] A study that used cardiac CT imaging to identify coronary artery calcium found that vascular age was increased in over 40% of patients, with an average increase of 15 years over the chronological age, also found that current CD4 count was associated with higher vascular age.[23] Atherosclerosis is an inflammatory process of the subintimal layer of the arterial wall in which lymphocytes and macrophages play a major role. The CD4+ type 1 T helper (Th1) lymphocyte is the predominant subtype of T cells in atherosclerotic plaques of humans.[24 25] Furthermore it has been demonstrated in a mouse model that CD4 cells play a pathogenic role in atherosclerosis.[26] Therefore ART-induced increase in CD4 count may contribute to the development of atherosclerosis in patients who are HIV-positive. Other studies have shown that low CD4 count (<350 cell/mL) is associated with higher rates of CVD or subclinical atherosclerosis. It may be that the association of CD4 with CVD is a U-shaped curve with low CD4 associated with acute inflammatory processes and high CD4 associated with chronic ongoing inflammatory processes.[23] However, in our study the association of CD4 with heart age may be mediated through components of the Framingham risk score, such as SBP or cholesterol.

### Risk prediction

Although some HIV-specific coronary heart disease (CHD) and CVD prediction algorithms have been proposed,[6 8] none have been externally validated by independent data and the Framingham risk score is still widely used.[10 27 28] Risk prediction in HIV-positive populations first focused on CHD,[7] but now the importance

of risk assessment for CVD has been recognised by guidelines.[5] D'Agostino summarised the state of CVD risk prediction as applied to HIV populations in a review article.[28] Studies in HIV populations have compared the degree of correlation of three traditional risk prediction algorithms, Framingham, Systematic Coronary Risk Evaluation (SCORE) and Prospective Cardiovascular Munster equations.[29 30 31] The estimation of relative effects of traditional risk factors on CVD outcomes appears similar between HIV-positive and HIV-negative individuals.[32] However, it may be that HIV-specific risk equations that include HIV-specific risk factors may perform better than existing algorithms because of potential differences in aetiology of CVD in the HIV population. For example, D-dimer, a marker of inflammation, has been found to be independently predictive of CVD events.[33] However, the Framingham risk score may partially capture inflammation since markers of inflammation have been found to be associated with a higher score in HIV patients compared with controls[34] and the score has been shown to correlate with the presence of subclinical atherosclerosis measured by carotid artery intima-media thickness in HIV-positive individuals.[35 36] Atherosclerosis may also be high in untreated patients, supporting a role of HIV infection itself as a risk factor.[37]

We used the Framingham risk score because it is based on readily available measures and widely used in clinical practice. Also, our focus was on factors that could be changed by lifestyle interventions, in particular smoking, blood pressure and cholesterol. Alternative scores have used different risk factors, for example, QRISK[38] includes ethnicity and family history of coronary heart disease which are not modifiable, body mass index (BMI), deprivation score, atrial fibrillation, rheumatoid arthritis and chronic renal disease. Although these risk factors are not entered in the Framingham risk score, they may be important in the evaluation of clinical risk. For example, renal insufficiency has a relatively high prevalence in HIV-infected black patients and may be an important contributor to risk in this population.[39]

The ability to accurately predict CVD risk is an essential element of this population's care. The Framingham equation for CHD predicted well in The Data Collection on Adverse events of Anti-HIV Drugs Study (D A D) in terms of discrimination, but tended to underestimate risk in smokers.[7] The Framingham risk scores may require recalibration to adjust for over or under prediction in the HIV population.[40 41] Factors unique to HIV, such as effects of different antiretroviral drugs, may influence the performance of standard risk prediction tools, as they may change CVD risk both through alterations of traditional risk factors and by contributing to inflammatory and immunological risk factors. D A D found both observed and predicted rates of myocardial infarction increased with time on ART implying that ART-induced changes in conventional risk factors at

least partially explained the increase in risk of MI.[7] The D A D predictive model for CVD tailored to patients with HIV included exposure to indinavir, boosted lopinavir and abacavir as well as the traditional Framingham risk factors.[8]

The HIV-HEART study found that not only were CVD risk factors high in the HIV population, but also that they are under-treated and therefore risk factor management of patients with HIV requires further improvement.[27] The European AIDS Clinical Society (EACS) guidelines on the prevention and management of metabolic disease in HIV[5] state that CVD risk should be assessed in HIV-positive individuals at regular intervals. They recommend lifestyle interventions should focus on counselling to stop smoking, modify diet and take regular exercise. A healthy diet, exercise and maintaining normal body weight tend to reduce dyslipidaemia.

## Smoking

Increased Framingham risk scores have been found in HIV-positive compared with negative controls, but this has mostly been due to higher prevalence of smoking rather than to higher cholesterol.[42] Smoking is the most important modifiable risk factor. A pilot study of a smoking cessation programme using counselling and nicotine replacement therapy in the Swiss HIV Cohort Study[43] found that implementing a smoking cessation programme was feasible in HIV-positive individuals. The DAD study found that the adjusted incidence rate ratio of CVD decreased from 2.32 within the first year of stopping smoking to 1.49 after greater than 3 years compared with those who never smoked. A recent study from Denmark estimated that HIV-positive individuals lose more years from smoking than from HIV infection.[19]

## Future work

In order for accurate CVD risk assessment to be carried out, HIV cohorts need to collect better information on Framingham risk factors and, in particular, on current smoking status of patients. Our study should be considered as a pilot study for assessing heart age. Ideally, this type of analysis would need to be rolled out into a much larger national cohort to properly gauge which factors predict heart age deviation. In addition, CVD events and death would allow us to test whether the Framingham equations are transferable to the UK HIV population. We need to assess the use of heart age as a communication tool for behavioural intervention. More research is required to investigate whether Framingham risk estimates accurately translate to actual CVD events and deaths or whether including HIV-specific risk factors in the CVD risk equations might result in a more accurate prediction tool for this population.

## Implications and conclusion

HIV-positive individuals in this cohort had a considerably increased risk of CVD compared with the ideal reference values. Our research, which showed that heart

age exceeds real age at all ages in men and above 40 years in women, implies that it is important to estimate CVD risk in HIV-positive individuals. The effect of smoking is to increase heart age on average by 8–17 years in men and 8–18 years in women depending on the mean real age. This indicates the importance of smoking cessation for prevention of CVD in this population. Furthermore, since the gap between heart age and real age increased as people get older, it is important to intervene in lifestyle factors such as smoking and obesity at young ages. Cardiovascular risk expressed in terms of an individual's heart age, rather than absolute risk of CVD, may have more impact on patients in programmes aiming to intervene in risk factors. Heart age may be a particularly useful tool in communicating risk to younger patients who are at low absolute risk of CVD. Tracking change in heart age, such as that due to smoking cessation, may provide strong motivation towards lifestyle changes.

**Author affiliations**
[1]School of Social and Community Medicine, University of Bristol, Bristol, UK
[2]HIV Service, North Bristol NHS Trust, Bristol, UK

**Acknowledgements** We thank all patients, Susan Allan and the team at Southmead Hospital, Bristol, UK.

**Contributors** MG and MM contributed to study design and interpretation of the analyses. TD analysed the data, compiled the tables and graphs and wrote the first draft of the paper. MM supervised statistical analyses. All authors contributed to the writing of the paper, edited, revised and approved the final version of the paper and had full access to all the data in the study. MM had the final responsibility for the decision to submit for publication and is the guarantor.

**Funding** Margaret May is funded by the Medical Research Council, UK (grant MR/J002380/1). Teri Davies is funded by the UK Medical Research Council as a PhD student in the Bristol Centre for Systems Biomedicine (BCSBmed) doctoral training centre. The views expressed in this manuscript are those of the researchers and not necessarily those of the Medical Research Council. The funders had no role in carrying out the study or in the decision to submit the manuscript for publication.

**Competing interests** None.

**Ethics approval** The project was approved by a Multicentre Research Ethics Committee and by local ethics committees.

**Provenance and peer review** Not commissioned; externally peer reviewed.

**Data sharing statement** No additional data are available.

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
