## [Reviewer comments · BMJ Open]

Some articles will have been accepted based in part or entirely on reviews undertaken for other BMJ Group journals. These will be reproduced where possible.

ARTICLE DETAILS

TITLE (PROVISIONAL)	Mind the gap: difference between Framingham heart age and real age increases with age in HIV-positive individuals: clinical cohort study
AUTHORS	May, Margaret; Davies, Teri-Louise; Gompels, Mark; Johnston, Sarah; Bovill, Begoña

VERSION 1 - REVIEW

REVIEWER	Abgrall, Sophie UPMC University Paris, INSERM
REVIEW RETURNED	12-Jun-2013

GENERAL COMMENTS	This article is an original work that evaluates differences in current age with the estimated heart age of the patients with ideal risk factors values by using the Framingham 10-year risk of CVD equations and risk factors associated with these differences. The results are that in male patients aged 45 years, the average heart age is 48 years for a male who does not smoke and 60 years for a male who smokes, and in female aged 44 years is 47 when non smoking and 59 when smoking. The differences increase with age, current CD4 count, ARV treatment with unsuppressed VL in smoker, and decrease when ART was initiated before 2003. Results are very well and clearly presented and clearly discussed. As the cohort did not have all information on all cardiovascular risk factors, the authors tried to implement different models to take into account potential biases. It is an interesting approach in cardiovascular prevention to help patients managing their preventable CV risks, such as smoking, obesity, diabetes mellitus, dyslipidemia. I think one of the main messages of the article is that smokers heart ages are on average 8 to 12 years older in males and 8 to 18 older in females than in non smoker males and females depending on the mean real age, and could be a little bit more emphasized in terms of prevention. As the authors did not have smoking status of the patients, they calculated the score twice, by evaluating it assuming that patients did not smoke, then by assuming patients smoked. Definition of smoking should be precised: is it can be " ever smoked", "current smoking", "smoking more than 5 or 10 cigarettes a day"...? Other cardiovascular risks are not detailed in the cohort such as genetic background (family cardiovascular history) and Body Mass Index, such factors do not enter in the Framingham score calculation, however they are important in the evaluation of the clinical risk. This
---

could be a little bit discussed.

The conclusion of the abstract stating that heart age deviation increases with age and CD4 is likely due to higher cholesterol in those with antiretroviral treatment can be a little rapid. It can also be due to an increase in BMI and lipodystrophy in patients with good immunovirological status on ARV.

Minor comments

Introduction:

Ref 10: I did not read in d'Agostino's article that the British Heart Foundation promoted the use of 'Heart Age' assessed in d'Agostino's article. Please give the reference of the British Heart Foundation guidelines.

Ref 12: please give a right reference to the Framingham heart study website.

A few list of preventable and non preventable cardiovascular risk factors should be given, as the Framingham score focuses only on some of them despite others can be important (such as family risk factors, renal insufficiency especially in HIV infected black patients who represents 30% of the cohort...)

Methods

Please give the definition of the smoking status used in the Framingham risk score (last cigarette within the last month?).

Results

Last sentence of the first paragraph: the median latest CD4 ... is in fact intermediate (and not totally comparable) between the CD4 of treated patients and CD4 of untreated patients

Discussion

We would like to have comparison with the general population. Despite not having age and sex matched HIV negative population, do the authors have any results of such heart age estimations within the general population in England (according to age and sex)? However, the published potential differences in effect of traditional risk factors on CVD between HIV infected and non infected population have been briefly discussed in the article.

How do the authors explain the results of table 3 with ART start date pre 2003 associated with a lower difference in heart age deviation?

Nevertheless, discussion section is very well written with potential biases plainly discussed, as are clinical implications of the article.

REVIEWER	Friis-Moller, Nina University of Copenhagen
REVIEW RETURNED	30-Jul-2013

THE STUDY	The study is entirely based on modeling and assumptions – no clinical or surrogate CVD outcomes are available or presented to confirm the assumptions. Incomplete CVD risk factors were available for the models, no recalibration has been attempted, no comparison of estimated CVD risk in HIV uninfected.
GENERAL COMMENTS	This paper examines the difference between estimated 'Framingham heart age' and real age for a British cohort of HIV positive individuals. The study adds potentially useful information to the field, but it also has some important limitations: Main comments:  1. The study is entirely based on modeling and assumptions – no clinical CVD outcomes are available or presented to confirm the assumptions based on applying the Framingham model to this cohort. Although other studies are referenced, which have explored the Framingham model in HIV positive individuals, this was a slightly different model (developed by Anderson et al) than the one used in the present study. 2. As discussed by the authors, it is recommended to recalibrate the Framingham equation to the particular population – this can be done using information on the CVD incidence in their population. This should be attempted, at least in sensitivity analyses 3. As acknowledged by the authors, only limited information on CVD risks is available – no data on smoking have been collected, which is one of the key CVD risk factors. 4. It is unclear which risk factors, that are driving the increasing gap between real age and predicted 'heart age' with increasing age – how much of the increasing gap may be inherent to the risk-function, and how much explained by worsening of CVD risk factors (dyslipidemia, hypertension, diabetes) in the HIV cohort with increasing age? For the latter, is this over and above age-related changes that are observed among HIV uninfected? It would be helpful to clarify this, and provide some reference to comparable analyses in HIV uninfected (if available). 5. The prevalence of core risk factors incl. diabetes should be included - preferably presented by gender and age groups

VERSION 1 – AUTHOR RESPONSE

Reviewer: Sophie ABGRALL
MD, PhD
INSERM U943, Paris; Avicenne Hospital, Bobigny;
France
No competing interests

1) This article is an original work that evaluates differences in current age with the estimated heart age of the patients with ideal risk factors values by using the Framingham 10-year risk of CVD equations and risk factors associated with these differences. The results are that in male patients aged 45 years, the average heart age is 48 years for a male who does not smoke and 60 years for a male who smokes, and in female aged 44 years is 47 when non smoking and 59 when smoking. The differences increase with age, current CD4 count, ARV treatment with unsuppressed VL in smoker, and decrease when ART was initiated before 2003.
Results are very well and clearly presented and clearly discussed.

RESPONSE: We thank the reviewer for this favourable comment.

2) As the cohort did not have all information on all cardiovascular risk factors, the authors tried to implement different models to take into account potential biases. It is an interesting approach in cardiovascular prevention to help patients managing their preventable CV risks, such as smoking, obesity, diabetes mellitus, dyslipidemia.

I think one of the main messages of the article is that smokers heart ages are on average 8 to 12 years older in males and 8 to 18 older in females than in non smoker males and females depending on the mean real age, and could be a little bit more emphasized in terms of prevention.

RESPONSE: We agree with the reviewer and have added to the Implications and conclusion section P16 “The effect of smoking is to increase heart age on average by 8 to 17 years in males and 8 to 18 years in females depending on the mean real age. This indicates the importance of smoking cessation for prevention of CVD in this population.” (Note we checked the figures using data from table 2 – it should be 8 to 17 for males and 8-18 for females)

3) As the authors did not have smoking status of the patients, they calculated the score twice, by evaluating it assuming that patients did not smoke, then by assuming patients smoked. Definition of smoking should be precised: is it can be “ ever smoked”, “current smoking”, “smoking more than 5 or 10 cigarettes a day”...?

RESPONSE: We have clarified that it is “current smoking (yes/no)” in the methods P 7.

4) Other cardiovascular risks are not detailed in the cohort such as genetic background (family cardiovascular history) and Body Mass Index, such factors do not enter in the Framingham score calculation, however they are important in the evaluation of the clinical risk. This could be a little bit discussed.

RESPONSE: We have added P14 “We used the Framingham risk score because it is based on readily available measures and widely used in clinical practice. Also, our focus was on factors that could be changed by lifestyle interventions, in particular smoking, blood pressure and cholesterol. Alternative scores have used different risk factors, for example, QRISK includes ethnicity and family history of coronary heart disease which are not modifiable, body mass index (BMI), deprivation score, atrial fibrillation, rheumatoid arthritis, and chronic renal disease. Although these risk factors are not entered in the Framingham risk score, they may be important in the evaluation of clinical risk.”

5) The conclusion of the abstract stating that heart age deviation increases with age and CD4 is likely due to higher cholesterol in those with antiretroviral treatment can be a little rapid. It can also be due to an increase in BMI and lipodystrophy in patients with good immunovirological status on ARV.

RESPONSE: We have removed this sentence from the conclusion of the abstract and have emphasised point 2) above instead that “heart age is very dependent on smoking status.”

Minor comments

Introduction:

6) Ref 10: I did not read in d’Agostino’s article that the British Heart Foundation promoted the use of ‘Heart Age’ assessed in d’Agostino’s article. Please give the reference of the British Heart Foundation guidelines.

RESPONSE: Apologies – we meant that the heart age calculator is based on the d’Agostino article. We have clarified this and changed the reference from the British Heart Foundation (as this is no

longer on the internet) to The Canadian Cardiovascular Society (which is current) (<http://www.ccsguidelineprograms.ca/index.php>) in the summary and on P3.

7) Ref 12: please give a right reference to the Framingham heart study website.

RESPONSE: Thank you – we have corrected this: Framingham Heart Study. General Cardiovascular Disease (10-year risk), 2013. Accessed 02/08/2013
<http://www.framinghamheartstudy.org/risk/gencardio.html>

8) A few list of preventable and non preventable cardiovascular risk factors should be given, as the Framingham score focuses only on some of them despite others can be important (such as family risk factors, renal insufficiency especially in HIV infected black patients who represents 30% of the cohort...)

RESPONSE: See answer to 4) above. We also added P14 “ For example, renal insufficiency has a relatively high prevalence in HIV infected black patients and may be an important contributor to risk in this population.”

Methods

9) Please give the definition of the smoking status used in the Framingham risk score (last cigarette within the last month?).

RESPONSE: See 3) above.

Results

10) Last sentence of the first paragraph: the median latest CD4 ... is in fact intermediate (and not totally comparable) between the CD4 of treated patients and CD4 of untreated patients

RESPONSE: Thank you – we have changed the wording on P8 to “was intermediate between”

Discussion

11) We would like to have comparison with the general population. Despite not having age and sex matched HIV negative population, do the authors have any results of such heart age estimations within the general population in England (according to age and sex)? However, the published potential differences in effect of traditional risk factors on CVD between HIV infected and non infected population have been briefly discussed in the article.

RESPONSE: As we explained on P11, we do not have access to a comparator HIV-ve population. Our focus is on comparing the HIV population with standardised ideal risk factors rather than with an actual population.

12) How do the authors explain the results of table 3 with ART start date pre 2003 associated with a lower difference in heart age deviation?

RESPONSE: We do not have an explanation for this finding.

13) Nevertheless, discussion section is very well written with potential biases plainly discussed, as are clinical implications of the article.

RESPONSE: We thank the reviewer for this favourable comment and her careful reading of the paper.

Reviewer: Nina Friis-Moller
University of Copenhagen

The study is entirely based on modeling and assumptions – no clinical or surrogate CVD outcomes are available or presented to confirm the assumptions. Incomplete CVD risk factors were available for the models, no recalibration has been attempted, no comparison of estimated CVD risk in HIV uninfected.

This paper examines the difference between estimated 'Framingham heart age' and real age for a British cohort of HIV positive individuals. The study adds potentially useful information to the field, but it also has some important limitations.

Main comments:

1. The study is entirely based on modeling and assumptions – no clinical CVD outcomes are available or presented to confirm the assumptions based on applying the Framingham model to this cohort. Although other studies are referenced, which have explored the Framingham model in HIV positive individuals, this was a slightly different model (developed by Anderson et al) than the one used in the present study.

RESPONSE: Our purpose was to promote heart age as a tool for communicating CVD risk to patients and to examine CVD risk factors in the Bristol HIV cohort to see how they impacted heart age. Therefore it is appropriate that it is partly based on clinical data and partly on modelling assumptions which are clearly stated throughout the paper.

2. As discussed by the authors, it is recommended to recalibrate the Framingham equation to the particular population – this can be done using information on the CVD incidence in their population. This should be attempted, at least in sensitivity analyses

RESPONSE: Our purpose was not to validate the Framingham risk score in our cohort. We do not have the data on CVD outcomes to recalibrate the Framingham risk score as explained on P11. Moreover, we specify on P16 that a larger cohort would be required to do this work and that it is future work that needs to be funded.

3. As acknowledged by the authors, only limited information on CVD risks is available – no data on smoking have been collected, which is one of the key CVD risk factors.

RESPONSE: We state on P11 that this is a major limitation of our work. Nevertheless, by using modelling assumptions we have shown the effect of smoking on heart age in this HIV+ve population.

4. It is unclear which risk factors, that are driving the increasing gap between real age and predicted 'heart age' with increasing age – how much of the increasing gap may be inherent to the risk-function, and how much explained by worsening of CVD risk factors (dyslipidemia, hypertension, diabetes) in the HIV cohort with increasing age? For the latter, is this over and above age-related changes that are observed among HIV uninfected? It would be helpful to clarify this, and provide some reference to comparable analyses in HIV uninfected (if available).

RESPONSE: We added a comment to the discussion on P12 "However, in our study, the widening of the gap between heart age and real age seen in older patients must be entirely driven by the risk factors included in the Framingham equation, namely diabetes, SBP, total and HDL cholesterol, being worse at older ages." Also see answer to reviewer 1 comment 11 above. There are no comparable analyses in available in HIV uninfected populations.

5. The prevalence of core risk factors incl. diabetes should be included - preferably presented by

gender and age groups

RESPONSE: We have examined this – see table below.

Sex Age category (years) No. Patients (%) Median total cholesterol (mg/dL) Median HDL cholesterol (mg/dL) Median systolic blood pressure (mmHg) Diabetes

N(%)

males 18-39 184(37%) 177.6 42.5 133 3 (0.16)

40-49 182(36%) 193 46.3 135.5 4 (2.2)

50-59 91(18%) 200.7 46.3 129 3 (3.3)

60- 46(9%) 189.1 46.3 131 7 (15.2)

females 18-39 126(51%) 185.3 54.0 121 1 (0.8)

40-49 83(34%) 193 54.0 126 3 (3.6)

50-59 27(11%) 212.3 61.8 127 3 (11.1)

60- 10(4%) 241.3 57.9 141.5 0 (0)

However, we do not think this table is very informative. We have added in the results P. 9 “In our study, prevalence of diabetes increased with age, as expected. Median total and HDL cholesterol were higher in men aged ≥ 40 years compared with younger men, and were also higher in women aged ≥ 50 years compared with younger women.”